# Prevalence and Molecular Characterization of *Salmonella* Isolated from Broiler Farms at the Tolima Region—Colombia

**DOI:** 10.3390/ani11040970

**Published:** 2021-03-31

**Authors:** Roy Rodríguez-Hernández, Johan F. Bernal, Jimmy F. Cifuentes, Luz Clemencia Fandiño, María P. Herrera-Sánchez, Iang Rondón-Barragán, Noel Verjan Garcia

**Affiliations:** 1Poultry Research Group, Faculty of Veterinary Medicine, University of Tolima, Altos the Santa Helena, A.A 546, Ibagué 730006299, Colombia; royrodriguezh@ut.edu.co (R.R.-H.); lfandino@ut.edu.co (L.C.F.); mpherreras@ut.edu.co (M.P.H.-S.); nverjang@ut.edu.co (N.V.G.); 2Investigación y Vigilancia Integrada de la Resistencia Antimicrobiana, Agrosavia, Km 14 Vía Mosquera—Bogotá CI Tibaitatá, Mosquera 250047, Colombia; jfbernal@agrosavia.co; 3Immunobiology and Pathogenesis Research Group, Faculty of Veterinary Medicine, University of Tolima, Altos the Santa Helena, A.A 546, Ibagué 730006299, Colombia; jimmytoci@gmail.com

**Keywords:** broilers, *Salmonella*, PFGE, prevalence, risk factors

## Abstract

**Simple Summary:**

*Salmonella* spp. is a major foodborne pathogen with a worldwide distribution that is responsible for salmonellosis in animals and humans. *Salmonella* contamination of poultry and derivative products occurs at different levels. The aim of this study was to estimate the prevalence of *Salmonella* in broilers farms and to conduct a phenotypic and molecular characterization of *Salmonella* isolates. The prevalence at the broiler farm level was 26.67%, and all isolates were found to belong to the serovar *Salmonella* Paratyphi B. These results suggest a common source of *Salmonella* contamination between broiler farms, presumably via feed, parent flocks or hatchery machines. *Salmonella* Paratyphi B is present in different segments of the poultry chain in the Tolima region. Additional studies are needed to identify the main source of *Salmonella* in broilers, chicken carcasses, and eggs commercialized in the Tolima region.

**Abstract:**

*Salmonella* is an important animal and human pathogen responsible for Salmonellosis, and it is frequently associated with the consumption of contaminated poultry products. The aim of this study was to estimate the prevalence of *Salmonella* in the poultry farms and to determine the genetic relationship. A total of 135 samples collected from fifteen broiler farms, including cloacal, feed, water, environmental and farm operator faeces samples were subjected to microbiological isolation. Molecular confirmation of *Salmonella* isolates was carried out by amplification of the *invA* gene, discrimination of d-tartrate-fermenting *Salmonella* isolates using multiplex PCR, and subsequently analysed by pulsed-field gel electrophoresis (PFGE). A survey questionnaire was conducted to identify potential risk factors for *Salmonella* presence in broiler farms. The prevalence of *Salmonella* at the farm level was 26.67%, and *Salmonella* isolates were serotyped as *S*. Paratyphi B and all isolates were d-tartrate-fermenting (dT+). PFGE showed three highly similar clusters and one significantly different *Salmonella* isolate. *S*. Paratyphi B continued to be present in different links of the poultry chain in the Tolima region, and identification of its main source is necessary to control its dissemination.

## 1. Introduction

Globally, foodborne diseases are common illnesses caused by consumption of contaminated food; nearly 550 million people become ill from diarrheal disease, including 220 million children [1]. *Salmonella* spp. is a major foodborne pathogen with worldwide distribution that is responsible for salmonellosis in animals and humans [2,3,4]. Consumption of *Salmonella*-contaminated food may result in human non-typhoidal salmonellosis, which is one of the leading causes of gastroenteritis in the world [5]. Human salmonellosis occurs mainly by ingestion of contaminated food from animal origin, frequently from poultry products such as eggs and raw chicken [6,7]. In the United States, *Salmonella* spp. is responsible for about 1.35 million infections, and 420 deaths every year, and in the European Union nearly 91,000 cases occur each year [3,8]. In Colombia, the National Health Institute (INS) in Bogota, reported 13,769 cases of foodborne diseases during 2018, of which 911 strains of *Salmonella* were isolated and a similar number of *Salmonella* spp. were obtained in 2015 and 2016. Of note, *S*. Typhimurium and *S*. Enteritidis continue being the most prevalent serovars [9].

*Salmonella* contamination of poultry and derived products may occur at different levels. Several sources of *Salmonella* infection/contamination have been described in primary production, including contaminated feed and water, asymptomatic birds, wild birds, rats and flies [10,11,12]. Second, in the post-production phase, during processing of chickens in abattoirs, contact with contaminated surfaces of cages during transport and market exhibition, among others, have been described as potential sources of *Salmonella* contamination [12,13]. Finally, direct contamination of poultry products by food handlers during cooking at formal and informal food vendors and homes cannot be excluded.

In Colombia, studies conducted in the poultry production chain in 2012 reported a 41.4% prevalence of *Salmonella* in broiler farms from Santander and Cundinamarca regions [14]. Later, the prevalence of *Salmonella* in broiler farms sampled in Santander in 2015 was only 2.8% [15]. In the Tolima, a prevalence of 33.33% *Salmonella* was reported in laying hen farms [12], likewise, a prevalence of 17.41% and 2.93% were reported in raw chicken meat and commercialized eggs in the same region, respectively [12,13]. In addition, our research group confirmed the presence of *Salmonella* spp. in about 10% (*n* = 110) of clinical cases of human gastroenteritis admitted in the period of August to December 2015 from local health care centres in Ibagué city [7]. Likewise, our studies established a clonal relationship between *S*. Enteritidis from poultry and human isolates, revealing the importance of *Salmonella* in public health and the need for robust studies with deep coverage that should address additional risks of contamination, the distribution and association with human clinical events. To increase the understanding of *Salmonella* epidemiology in the Tolima region, for this purpose the aim of this study was to estimate the prevalence of *Salmonella* in poultry farms and to determine the genetic relationship.

## 2. Materials and Methods

### 2.1. Study Population

The Tolima district is located between the central and eastern mountains of the Colombian Andes, and it has 42 municipalities with approximately 4 million broilers at 90 farms. The Tolima region, together with Cundinamarca, Huila and Boyacá, represent the largest broiler producer regions in Colombia.

### 2.2. Study Design

A cross-sectional study was designed to estimate the prevalence of *Salmonella* spp. in broiler farms. The sampled farms were selected by convenience. We chose a non-probabilistic method because the study depended on the farm owner’s decision to participate in the study. Thus, only 15 commercial broiler farms from three different companies in the Tolima region allowed us to conduct confidential sampling. Two poultry houses for each farm were sampled for *Salmonella* detection using different samples types. All poultry houses sampled were surrounded by a natural environment, and the bird density ranged from 7 to 9 hens/m^2^. The facilities structures were made in bamboo or metal and the floor was soil or concrete. All farms were found to store the feed bags inside the poultry house. Human faecal samples were voluntarily provided by poultry operators involved in the study through informed consent, based on Resolution 8430-1993 from the Colombian Healthcare department. This study was approved by a Central Research office committee at the University of Tolima, and the approval number was 130214.

### 2.3. Sample Collection

A total of 135 samples were collected from fifteen broiler farms between the period 2015–2016. The samples included cloacal swab pools (*n* = 75), feed samples (*n* = 15) water samples (*n* = 15), boots swab samples (*n* = 15) and faecal samples (*n* = 15) provided by farm operators. Cloacal swabs were obtained from chickens selected randomly in each house, making 5 pools of 10 birds per flock (50 bird/flock) and a total of 750 birds were sampled. The age of the sampled chickens ranged between 12 to 42 days-old. The feed samples were taken from newly opened feed bags at the moment of the sampling and consisted of 250 g of feed collected in sterile bags (Nasco^®^, Fort Atkinson, WI, USA). The water samples consisted of 250 mL collected in Whirl-Pak™ Standard Sample Bags (Nasco^®^, Fort Atkinson, WI, USA) taken from the last drinker of the poultry house. The environmental samples consisted of boot swabs obtained by covering rubber boots with sterile cotton covers, which were collected after walked for 30 min in a zigzag pattern over the length of the chicken house [16,17]. The faecal samples were obtained from farm operators that previously have given their consent. Samples were transported under refrigeration conditions (4 °C) to the Laboratory of Veterinary Diagnosis, University of Tolima, Ibagué, Colombia, and processed within 12 h for *Salmonella* isolation.

All experimental procedures followed the guidelines from the Bioethics committee of the Central Research Office from University of Tolima based on the Law 84/1989 and Resolution 8430/1993 of Colombia government, and the Guide for the Care and Use of Agricultural Animals in Research and Teaching [18].

### 2.4. Salmonella Isolation and Identification

Microbiological isolation of *Salmonella* spp. followed specific standard procedures ISO/6579: 2002/AMD1: 2007, the Colombian Technical Standard (NTC) 4574, the standard issued by the authority Instituto Colombiano Agropecuario (ICA). Briefly, feed and water samples were pre-enriched 1:10 in buffer peptone water (BPW) (Oxoid, Basingstoke, Hampshire, UK); the cloacal swabs pool and boot swabs were mixed with 25 mL of BPW. All samples were incubated at 37 °C ± 1 °C for 24 h. Then, 0.1 mL of pre-enrichment medium was inoculated into 10 mL of Rappaport Vassiliadis broth (Oxoid, Basingstoke, Hampshire, UK) and then it was incubated at 41.5 °C ± 1.0 °C for 24 h. A second aliquot (0.1 mL) of pre-enriched medium was inoculated into 10 mL Tetrathionate broth (Oxoid, Basingstoke, Hampshire, UK) and incubated at 37 °C ± 1 °C for 24 h. The faecal samples from farm workers were directly inoculated in Rappaport Vassiliadis and Tetrathionate broth and incubated at 41.5 °C ± 1.0 °C for 24 h. Then, an aliquot of Rappaport Vassiliadis and Tetrathionate cultures were inoculated into Xylose Lysine Deoxycholate agar (XLD) (Merck, Darmstadt, Germany) and MacConkey agar (Oxoid, Basingstoke, Hampshire, UK) selective solid media and incubated at 37 °C ± 1 °C for 24–48 h. Suspicious *Salmonella* colonies were confirmed by culture in Xylose Lysine Tergitol-4 (XLT4) agar (Merck, Darmstadt, Germany) at 35 °C ± 1 °C for 18–24 h and Rambach agar (Merck, Darmstadt, Germany) at 37 °C for 24–48 h. Subsequently, the *Salmonella* colonies were tested for agglutination with Poly A-I and Vi antiserum (Difco^®^ 222641; Becton Dickinson and Co, Sparks, MD, USA). Finally, the *Salmonella* isolates were biochemically characterized using the API-20E^®^ (BioMérieux’s, Marcy l’Etoile, France) enteric identification system. Farms were identified as positive for *Salmonella* when at least one sample was positive in the bacteriological culture. The *Salmonella* prevalence was calculated by dividing the number of farms classified as positive by the total number of sampled farms.

### 2.5. Molecular Confirmation of Salmonella

*Salmonella* isolates were confirmed by amplifying the *invA* gene, which is conserved in *Salmonella* serovars [19].

Genomic DNA (gDNA) was extracted from all *Salmonella* isolates using the Invisorb^®^ Spin Universal Kit (Stratec, Berlin, Germany) and a fragment of the *invA* gene was amplified by using the forward (5′-GTGAAATTATCGCCACGTTCGGGCAA-3′) and reverse 5′-(TCATCGCACCGTCAAAGGAACC-3′) primers [20]. Endpoint PCR was carried out in a total volume of 25 µL, containing 1 µL of gDNA template, 1 µL of forward primer, 1 µL of reverse primer, 1 µL of Taq polymerase, 2.5 µL of buffer 10X, 2 µL of MgCl_2_, 2.5 µL dNTP and 14 µL of nuclease free water. PCR was performed in a BIO-RAD T100™ thermal cycler (Bio-Rad, Hercules, CA, USA) with the following conditions—denaturation of 3 min at 95 °C, 35 cycles of amplification with 30 s at 95 °C (denaturation), 30 s at 55 °C (annealing), and 90 s at 72 °C (extension), followed by 5 min at 72 °C for final extension and then resolved by electrophoresis on 2% agarose gel. The reaction products were stained with HydraGreen™ (ACTGene, Piscataway, NJ, USA) and visualized under the UV light by using a trans-illumination system ENDURO^TM^ GDS (Labnet International, Inc, Woodbridge, NJ, USA).

### 2.6. Salmonella Serotyping

All *Salmonella* isolates from broilers farms were serotyped following the White–Kauffmann–Le Minor scheme for O and H antigens by using commercial antisera (Difco, Becton Dickinson and Co., Sparks, MD, USA). The serovars of *Salmonella* are based on the nomenclature described by the Judicial Commission of the International Committee on Systematics of Prokaryotes. The serotyping test was carried out at the National Laboratory of Veterinary Diagnosis of ICA (Bogotá, Colombia).

### 2.7. Molecular Characterization by Pulse Field Gel Electrophoresis (PFGE)

*Salmonella* isolates were analysed by PFGE using CHEF-DR III equipment (Bio-Rad, Hercules, CA, USA) at the National Reference PFGE Laboratory of Colombian National Health Institute (INS), following the PulseNet protocol [21,22]. Briefly, the gDNA from each *Salmonella* isolate was released into agarose plug gels and digested with the *XbaI* restriction enzyme (Promega, Madison, WI, USA). The DNA fragments were separated on a PFGE-certified 1% agarose gel (Bio-Rad, Hercules, CA, USA) with 0.5X Tris-borate-EDTA running buffer for 17 h at 6 V/cm with increasing pulse times. The *XbaI*-digested *Salmonella* Braenderup H9812 DNA was used as a reference and size standard. The gels were stained with ethidium bromide and analysed using the Gel Compare II^®^ software (Applied Maths, Sint-Martens-Latem, Belgium). The similarity was calculated by the Dice coefficient, and a dendrogram was constructed by cluster analysis using the unweighted pair group method with arithmetic mean (UPGMA). A band position tolerance of 1.5% was used for analysing the PFGE fingerprints.

### 2.8. Multiplex PCR for the Discrimination of dT+ and dT− Salmonella Isolates

For the discrimination of dT+ and dT− *Salmonella* isolates, a fragment of 290 bp was amplified by using primers 166 (5′-GTAAGGGTAATGGGTTCC-3′) and 167 (5′-CACATTATTCGCTCAATGGAG-3′) [23]. In addition, for the identification of the genus, primers ST11 (5′-AGCCAACCATTGCTAAATTGGCGCA-3′) and ST15 (5′-GGTAGAAATTCCCAGCGGGTACTG-3′) were used [24]. The PCR was carried out in a total volume of 25 μL, composed of 12.8 μL of distilled-deionized water, 5 μL of 5X green GoTaq^®^ Flexi Buffer (Promega, Madison, WI, USA), 2 μL of dNTPs (1.5 mM) (Invitrogen, Carlsbad, CA, USA), 1 μL of each primer (166 and 167) (10 pmol/μL), 0.5 μL of each primer (ST11 and ST15) (10 pmol/μL), 1 μL MgCl2 (25 mM), 0.125 μL of GoTaq^®^ Flexi DNA polymerase (Promega, Madison, WI, USA) and 1 μL of the gDNA as template. The amplification was carried out in a ProFlex PCR System (Applied Biosystems, Carlsbad, CA, USA) with an initial denaturation step at 95 °C for 3 min, followed by 35 cycles of denaturation at 95 °C for 30 s, annealing at 55 °C for 30 s, extension at 72 °C for 30 s and a last step of final extension at 72 °C for 5 min. Amplicons were revealed on 1% agarose gel by electrophoresis (PowerPac™ HC, Bio-Rad, Bio-Rad, Hercules, CA, USA) using GeneRuler 100 bp DNA Ladder (Thermo Fisher Scientific, Waltham, MA, USA). The gel was stained with HydraGreen™ (ACTGene, Piscataway, NJ, USA) and visualized under the UV light, using the ENDURO^TM^ GDS gel documentation system (Labnet International, Inc, Woodbridge, NJ, USA).

## 3. Results

### 3.1. Prevalence of Salmonella

A total of 15 farms were sampled from which four farms were found positive to *Salmonella* spp., which represents a prevalence of 26.67% (4/15). From all samples analysed (135), 17.78% were positive for *Salmonella* isolation (24/135) and the microorganism was isolated most frequently from cloacal swab samples (79.17%, *n* = 19), followed by the boot swabs samples (16.67%, *n* = 4) and feed samples (4.16%, *n* = 1). The sampled house farm facilities only had two types of structure—bamboo or metal, they also had a floor in concrete or soil. However, approximately 75% of the positive farms were characterized by facilities constructed from a bamboo and soil floor, and only 25% of positive farms had metallic and concrete floor facilities. All faecal samples provided by farm operators were negative for the presence of *Salmonella*.

### 3.2. Molecular Confirmation of Salmonella Genus by PCR

All *Salmonella* isolates were confirmed by PCR amplification of a 284 bp DNA fragment of the invasion gene *invA* (Figure 1).

### 3.3. Salmonella Serovars

The 24 *Salmonella* isolates were subjected to the White–Kauffmann–Le minor serotyping scheme and all isolates were identified as *Salmonella* Paratyphi B.

### 3.4. PFGE Analysis

Twenty-four *Salmonella* isolates were analysed by PFGE and showed three highly similar clusters (>89.9%) and one significantly different *Salmonella* isolate (Figure 2). Cluster I was composed of seven isolates from cloacal swabs and environmental swabs. Although five isolates showed 100% Dice similarity index, there were two isolates (UT2-10, 11) slightly different (one additional band) from the previous ones (Dice index of 97.1%). The cluster II was composed of six isolates obtained also from cloacal swabs and boot swabs and showed one *Salmonella* isolate slightly different (UT2-21) with a 99% Dice similarity index. The cluster III (Dice index of 100%) was composed of ten isolates that were obtained mostly from cloacal swabs but also from boots swabs and feed samples. Clusters I and II showed 92% Dice similarity index, and together showed 89.9% Dice similarity index with Cluster III. Finally, the isolate UT2-1 formed an independent pattern and showed only 10.2% Dice similarity index with previous clusters.

### 3.5. Molecular Discrimination of dT+ and dT− Salmonella Isolates

All *Salmonella* Paratyphi B (*n* = 24) isolates were d-tartrate-fermenting isolates (dT+) confirmed by amplification of a 290 bp fragment from the putative cation transporter STM 3356 gene (Figure 3).

## 4. Discussion

This study for the first time estimated a 26.67% prevalence of *Salmonella* in broiler farms sampled in Tolima region, Colombia. This result is significantly lower than the 41% prevalence previously reported in broiler farms from Santander and Cundinamarca regions in 2012 [14] and 40.5% median prevalence of *Salmonella* in broiler production estimated worldwide [25]. Since the prevalence of *Salmonella* in broiler carcasses marketed in Ibague city was 17.4% in 2014 and *S*. Paratyphi B was the serotype most frequently (36.17%) [26] isolated, together the data suggest the dissemination of *S*. Paratyphi B from poultry farm to consumer. *Salmonella* Paratyphi B variant java has been described as a reservoir of complex antimicrobial resistance profiles and diverse mobile genetic elements in poultry production in Colombia, and this variant was traced from farm to retails stores, but *Salmonella* Paratyphi B ST28 could have an amplified action range in human hosts in Latin America [27,28]. In addition, positive farms exhibit a common rudimentary production facility constructed from bamboo, which represents a possible additional risk because of its limited disinfection options and it could not represent an optimal physical barrier to avoid entrance of another natural reservoir [12]. These results show that the presence of *Salmonella* in poultry production could be dynamic in time and space [29,30], and health authorities must promote continued strengthening of integrated surveillance and guarantee *Salmonella* to be part of the local and national foodborne agenda, as a priority in public health. *Salmonella* has more than 2600 serovars, based on 46 O antigens and 114 H antigens [31] which may differ in their ability to colonise the host and persist in the environment. In our study, all isolates belonged to *Salmonella* Paratyphi B serovar which is recognized to have two variants, fermenting tartrate (dT+) and non-fermenting tartrate (dT−) [32]. These two variants have very different pathogenic characteristics—*S*. Paratyphi B (dT−) variant could produce invasive infections and paratyphoid fever with life threatening infection in humans [33]. However, in this study, *S*. Paratyphi B isolates were found to be tartrate (dT+) fermenting variants that are associated with acute diarrheic disease (self-controlling) [33,34]. *S*. Paratyphi B (dT+) has animal reservoirs that could be the source of infection [35]. On the other hand, this variant was the most prevalent in broiler flocks in Belgium (12.3%) and Germany (10.8%) [36]. However, the result of this study contrasts with the work done by Rodriguez et al. [26], where at least 14 different serovars were isolated from broiler carcasses (47/270) marketed in the capital of Tolima region. As has been discussed in previous works, the results strengthen the hypothesis that contamination of broiler carcasses with a number of potentially pathogenic serovars of *Salmonella* might be occurring. Some authors have reported *Salmonella* carcase contamination during handling and processing in abattoirs, transport and exhibition at the market place, and suggest poor hygienic and disinfection practices at those processes and steps of the poultry chain [37,38,39]. In our study, *Salmonella* Paratyphi B was isolated from one feed sample (0.74%), andalthough this frequency is very low, it raised concerns regarding the quality of the feed, since this may constitute an efficient mechanism of widespread contamination of poultry flocks with *Salmonella*. In line, other serovars such as *Salmonella* Shannon were also isolated from feed samples (28.57%) obtained in laying hen farms in this region of Colombia [12]. On the other hand, 96% (19/24) of *S*. Paratyphi B recovered was isolated from cloacal and environmental samples. These results could represent diverse contamination sources ranging from possible contaminated parental flocks or the hatchery machines and then contamination of flock to flock populations, to environmental contamination factors as poor bio-security practices, residual contamination (dust), inefficient flaming process, litter recycling, other biological reservoirs (pets or pests), depopulation time and sanitation [40,41,42].

Previous studies have reported that *S*. Paratyphi B isolated from poultry products in Colombia [14,43] have remarkable antibiotic resistance, and 22/24 *Salmonella* Paratyphi B isolates in this study were resistant to three antibiotic classes, penicillins (ampicillin), aminoglycosides (streptomycin), and cephalosporins (ceftriaxone and ceftazidime), that may have an impact in the treatments of human infections [44]. Furthermore, antimicrobial resistance constitutes a global trend and international food trading restrictions could be pressuring drivers to promote control of *Salmonella* in Colombia [33,45,46]. In the Netherlands and Germany, this variant was reported in isolates from poultry which showed multidrug-resistance (MDR), causing the disease in humans [32,47].

All isolates were grouped into three closely related PFGE clusters but one isolate had a very different PFGE pattern. These results indicate a possible common source of contamination with *Salmonella* between different farms, presumably water, contaminated feed, parent flocks or cross-contamination with environmental sources [28,31], based on the wide distribution of those clusters identified in broiler farms and high genetic relation (SI = 0.899) between them [48,49], and additional local studies suggesting high heterogeneity in reported PFGE patterns in a specific geographical distribution [14,48]. The results suggest an association between the homogeneity in PFGE patterns obtained in this study with a common contaminating source. However, contamination of broiler production systems with a single genetic related serovar could mean a unified plan of hygiene and disinfection control actions for all positive production systems sampled in the Tolima region, which is improbable.

## 5. Conclusions

*Salmonella* Paratyphi B was found as the predominant serovar present in broiler farms sampled in the Tolima region. Molecular characterization (PFGE) identified that those *Salmonella* Paratyphi B isolates presented high genetic relatedness, indicating a possible common contamination source across different farms. Despite this, *S*. Paratyphi B dT+ is not related to acute human illness in Colombia, however, being a successful reservoir of antimicrobial resistance, it is important for public health. Additional studies are needed to clarify the main source of *Salmonella* observed in the Tolima poultry industry, since *S*. Paratyphi B is a common serovar isolated not only from broiler farms, but also from marketed chicken carcasses, and eggs commercialized in the Tolima region, and it could represent a starting point. This study highlights the need to establish local or national monitoring programs to generate longitudinal data of pathogen dynamics in animal production in Colombia. By integrating animal data to food and clinical data, it would be possible to generate bio-contention plans and actions to early identify emerging pathogens of interest in animal and human health.

## Figures and Tables

**Figure 1 animals-11-00970-f001:**
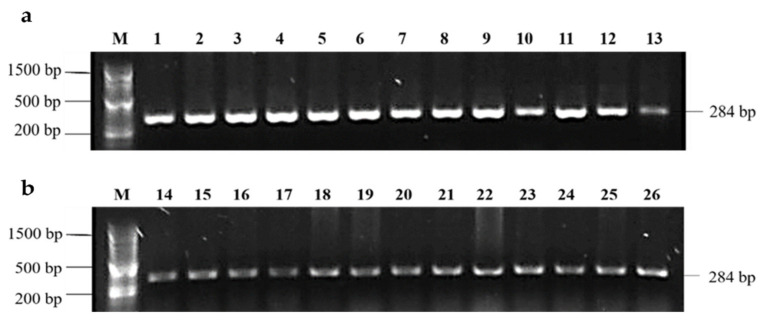
PCR amplification of 284 bp DNA fragment from the invasion gene *invA* of *Salmonella* isolated from broilers farms in Tolima, Colombia. M—100 bp marker; (**a**) lanes 1–12 and (**b**) lanes 14–25—*Salmonella* isolates from Tolima broiler farms; lane 13 and 26—positive control *Salmonella* Enteritidis ATCC 13076.

**Figure 2 animals-11-00970-f002:**
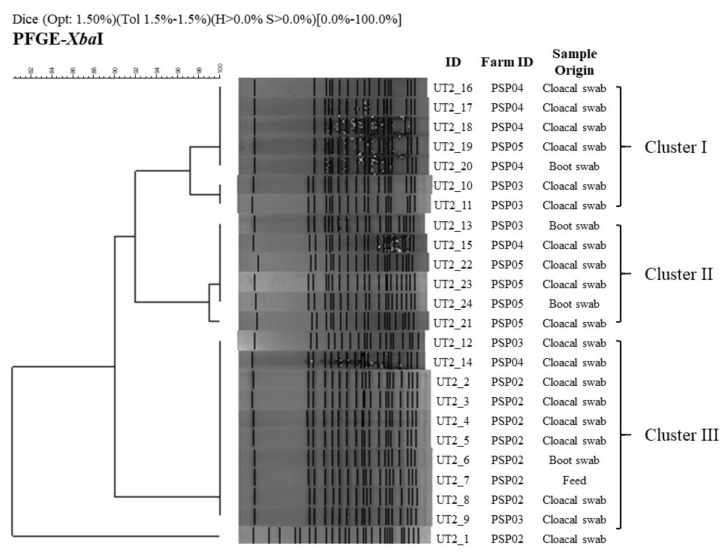
Macrorestriction patterns of 24 *Salmonella* Paratyphi B generated by pulsed-field gel electrophoresis (PFGE) of *XbaI*-digested genomic DNA. A similarity analysis was performed using the Dice coefficient, and the dendrogram was generated by the unweighted pair group method with arithmetic averages using the Gel Compare II software (Applied Maths, Sint-Martens-Latem, Belgium). Lanes UT2_1 to UT2_24 correspond to the genomic DNA fingerprinting from each *Salmonella* isolate from the broiler farms in the Tolima region collected from year 2015 to 2016.

**Figure 3 animals-11-00970-f003:**
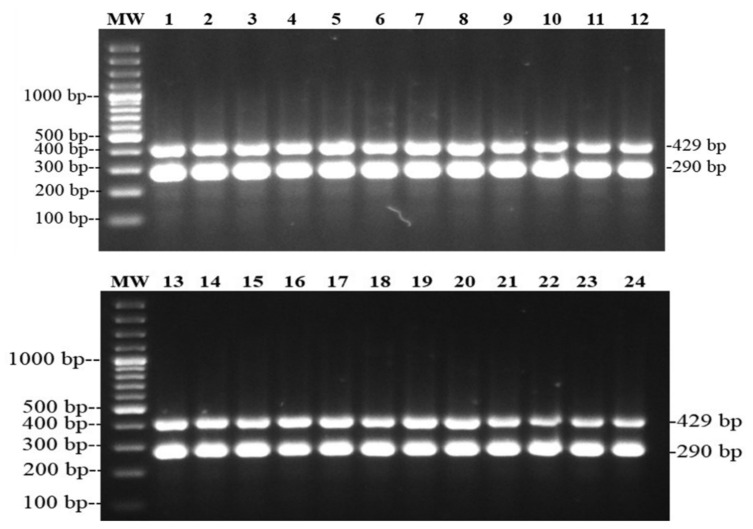
PCR amplification of 290 bp of the putative cation transporter STM3356 gene identifying dT+ *Salmonella* isolated from broilers farms in Tolima, Colombia. MW—GeneRuler 100 bp Plus DNA Ladder ((Thermo Fisher Scientific, Waltham, MA, USA); lanes 1–24: *Salmonella* Paratyphi B isolates from Tolima broiler farms. The 429 bp band corresponds to the genus specific *Salmonella* control.

## Data Availability

Not applicable.

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
