# Peer review of "Prevalence and Molecular Characterization of Salmonella Isolated from Broiler Farms at the Tolima Region—Colombia"

_animals, 2021, doi:10.3390/ani11040970_

Round 1

Reviewer 1 Report

The authors have generally addressed most of my point-by-point concerns. However, I am stuck on problems with Table 2 and I cannot for the life of me figure out how they derived the relative risks they present based on all 4/4 Salmonella positive farms all having all four of the risk factors in question (all of which are farm level risk factors). First of all, since all four positive farms have the 'exposures' in question, then the only thing that can vary is the negative farms status with respect to the exposures. However, we are not given a column with those data; but, presumably they vary somewhere between 0/11 farms (total = 15 farms, 4 were positive) and 11/11 farms. Regardless, since no 'exposure-negative' farms ever had Salmonella based on that table, the RR (or odds ratios) are always 'infinite'. Even if I artificially allowed one positive farm to be risk factor negative, I could never achieve the confidence intervals and incredibly low p-values as stated in the table.  The authors must either pull the table out and throw it away, or demonstrate to this reviewer that they know how to calculate a relative risk and show the data involved therein. Given the very few farms involved in this study (15) the number of positives = 4, and the fact that all four had all four risk factors (4/4 in column labeled proportion of positive farms) then the RR are wrong. If the authors wish to keep these data then present the proportion of negative farms positive for the four risk factors, but be aware that the incalculable/infinite RR/OR make these sort of silly.

Author Response

Response to Reviewer 1 Comments

Point 1: The authors have generally addressed most of my point-by-point concerns. However, I am stuck on problems with Table 2 and I cannot for the life of me figure out how they derived the relative risks they present based on all 4/4 Salmonella positive farms all having all four of the risk factors in question (all of which are farm level risk factors). First of all, since all four positive farms have the 'exposures' in question, then the only thing that can vary is the negative farms status with respect to the exposures. However, we are not given a column with those data; but, presumably they vary somewhere between 0/11 farms (total = 15 farms, 4 were positive) and 11/11 farms. Regardless, since no 'exposure-negative' farms ever had Salmonella based on that table, the RR (or odds ratios) are always 'infinite'. Even if I artificially allowed one positive farm to be risk factor negative, I could never achieve the confidence intervals and incredibly low p-values as stated in the table.  The authors must either pull the table out and throw it away, or demonstrate to this reviewer that they know how to calculate a relative risk and show the data involved therein. Given the very few farms involved in this study (15) the number of positives = 4, and the fact that all four had all four risk factors (4/4 in column labeled proportion of positive farms) then the RR are wrong. If the authors wish to keep these data then present the proportion of negative farms positive for the four risk factors, but be aware that the incalculable/infinite RR/OR make these sort of silly.

Response: Thank you for your advice, since there are very few farms involved in this study, the information from the survey was not relevant and did not affect the discussion and conclusions, and following your suggestion, we prefer not to include the information from the epidemiological survey and Tables 1 and 2 in the manuscript.

Reviewer 2 Report

Please correct the following points in the manuscript: 

- Line 74. Insert a presentation for the objective. "For this purpose the aim of this study was to..."

 - Line 265. Salmonella serovars

- Result section: 
I can't find where it has been described how the ODDS of the risk factors has been analyzed.

Author Response

Response to Reviewer 2 Comments

Point 1: Line 74. Insert a presentation for the objective. "For this purpose the aim of this study was to..."

Response 1: The text has been rephrased and have included “For this purpose the aim of this study was to..."

 Point 2:  Line 265. Salmonella serovars

Response 2: We have made a respective change in the manuscript.

Point 3:  Result section: 
I can't find where it has been described how the ODDS of the risk factors has been analyzed.

Response 3: Thank you for your comment, since there are very few farms involved in this study, the information from the survey was not relevant and did not affect the discussion and conclusions, and following your suggestion from reviewer n ° 1, we prefer not to include the information from the epidemiological survey and Tables 1 and 2 in the manuscript.

This manuscript is a resubmission of an earlier submission. The following is a list of the peer review reports and author responses from that submission.

Round 1

Reviewer 1 Report

The article by Rodriguez-Hernandez et al. describes characteristics of Salmonella enterica (dominated entirely by serotype Paratyphi B d-tartrate-fermenting) among poultry isolates of differing stages of production in the Tolima region of Colombia. There are differing strains by Pulse Type, but clearly this serotype has found a suitable niche to spread and propagate within the poultry sector of this region and across farms. Broiler hatchery as a single source becomes highly suspected, though feed is mentioned as the authors primary hypothesis. The comment was made in the summary and abstract about ‘recirculation’, which begs further discussion for what is normally a linear and unidirectional production flow. The sampling of cloaca and environment in broiler farms does not readily confirm (or, reject) the stated hypothesis of recirculation so this should be made with some caveats and caution. In addition, the work is cross-sectional and not longitudinal and thus lacks the temporal aspects that would support ‘recirculation’ which infers time and flow. The work is original and adds support to previously published data from other regions of the country concerning S. Paratyphi B dominance in the poultry sector. I would suggest the authors spend a little time in the introduction and discussion explaining how this variant of Paratyphi B differs from S. Typhi and Paratyphi A, B and C, which are normally assumed to be human-to-human circulation only (as per US CDC), and why this distinction is important as it relates to potential for human foodborne illness. Some descriptions of the relative magnitude of Paratyphi B in the attribution of salmonellosis among humans in Colombia might also be warranted (they mention Typhimurium and Enteritidis). IN some countries, the dominant poultry serotype (e.g., some sequence types of S. Kentucky) have little relevance to human cases.

There are multiple minor typographical errors and mistakes throughout. I have only identified those appearing in Summary and abstract as examples here (immediately below). I suggest the authors simply run the article through Microsoft Word using the UK English language setting and the spellchecker and grammar checker and they will readily identify the rest of these minor errors and quickly fix them. Otherwise, the paper is mostly well-written and easy to follow.

Line 15: …may occur…

Line 23: …responsible for salmonellosis… (not a proper noun)

Line 24: …associated with…

Line 25: The aims of the study were…

Line 25: broiler farms in the Tolima region

Line 25: … and to identify…

Line 31: … risk factors for…

etc....

Line 40: start with ‘Globally, foodborne diseases…”

Lines 87-90: This is inconsistent with lines 26-27.

Lines 101-104: Which of these laws/regulations pertained to the collection of voluntarily provided human fecal materials and their analysis and reporting? An institutional review board number is usually required (there should be a number/approval issued by the university Central Research Office).

Lines 183-193: Again, in most jurisdictions there would be a requirement for stating an approval (or, exemption) for administration, analysis, and reporting of survey questions of individuals including statements of informed consent. Please clarify those here.

Line 197: Please use the word ‘strain’ carefully. Perhaps you mean isolate here, unless these have been determined to all be non-clonal?

Lines 198-199: what is the farm-level 95% confidence interval? Not sure what this means as stated? My quick calculation in Stata v 16.1 shows an exact 95% CI that is very wide:

. cii proportions 15 4, exact

                                                         -- Binomial Exact --

    Variable |        Obs  Proportion    Std. Err.       [95% Conf. Interval]

-------------+---------------------------------------------------------------

             |         15    .2666667    .1141798        .0778715    .5510032

Please confirm what the authors intended to say here.

Lines 199-204: Please state explicitly if these prevalence proportions refer to only those farms / flocks in which Salmonella were recovered or all farms?

Lines 201-202: as opposed to what other types of structures that were negative?

Lines 202-204: again, as compared to what?

Line 21: Change to Lane 13 & 26: positive control (not the range from 13-26).

Table 1: UT2-7: Please confirm if the feed samples were taken from the storage area or in the poultry house itself where they could be contaminated by the birds? Not clear from Line 88 if the samples were from ‘hoppers’ and free from potential contamination and thus introduced, or from the feeders in the houses?

Lines 219-221: technically speaking, the description is one of ‘phenotype’ d-tartrate-fermenting (dT+), but the method was molecular via PCR. While this is an accepted approach, the language is not specific.

Figure 2: Where is the positive control for this gene? Even including a Paratyphi B strains previously characterized by sequencing to have this gene would have been helpful.

Table 2: Be consistent with decimals versus commas between text and table. Suggest decimals (.). I am also not clear about the approach taken to calculate the odds ratios. Was this farm-level positivity (1/0) as 4/15? In which case there would be very limited power to detect these differences. I would suggest including a column with the proportion of positive farms that engaged in this practice and assume the practice would be almost non-existent on the negative farms?

What as the depopulation time units (hours?) Use only two significant digits for lower 95% CI. The p-values are also suspicious. If the last p-value was truly 0.05 then the lower 95% CI would typically be: 1.0, not 1.15.

Lines 234-247: The key part missing from this results description is the farm. The source (boot, cloaca, feed, etc.) are less important than knowing if these strains break out by farm or hatchery source, or feed source. Were these all from a single production company?

Figure 3: again, there might have been some value in including a highly characterized S. Paratyphi from the Colombian collection if such a strain existed and had been sequenced as a referent. Not essential here as positive control is not essential as it is in PCR.

Line 256: “At the farm level, not bird level”, please clarify here.

Lines 256-262: The comparisons to carcasses are ‘apples to oranges’ since the comparison of famr-level prevalence of Salmonella (a single positive sample means the farm is positive) does not translate readily to bird- or carcass-level prevalence. Likewise, it is important to know which comparison is being made ‘worldwide’ at 40.5%.

Line 265-267: The connection to antimicrobial resistance is not clear? These strains were not characterized or at least not reported here.

Lines 279-282: This is the first we learn that data from these isolates have been reported elsewhere already? I looked up the paper in Vet. World and see the paper reports on AMR profiles of these 24 isolates along with another batch of different serotypes. Importantly, these isolates appear to be MDR and include third generation cephalosporin resistance, probably encoded by blaCMY-2 but also potentially with blaCTX-M (though the connection of data points from Table 2 and Figure 1 in that paper are not clear). The dendrogram (of the 24 S. Paratyphi) in the Vet World paper is also remarkably close to the PFGE dendrogram in this submission, though performed using different molecular techniques! Consistency is a good thing!

Lines 285-307: Very good discussion section!

Lines 308-320: Interesting discussion but I do not think it leads to a conclusion, especially since all the isolates arose from just four farms.

Line 322: I am not convinced the authors have proven with these data that the serovar was ‘recirculating’ on these broiler farms. The alternative hypotheses such as a single source from a broiler hatchery or feedmill has not been ruled out and that is not ‘recirculation’.

Lines 343-355 were left as default, but they need to be completed especially for the human subjects’ components. No mention was made of those samples, so I assume they were negative for Salmonella?

Reviewer 2 Report

General comments:

Serovar or Serotype?  For some time now, the accepted situation is that "serotype" is a verb and that the noun is "serovar" - see Lapage, S.P., Sneath, P.H.A., Lessel, E.F., Skerman, V.B.D., Seeliger, H.P.R., Clark, W.A., 1976. International Code of Nomenclature of Bacteria. American Society for Microbiology, Washington.  Please move forward and adopt the noun "serovar".

As far as I know, the serovar that has been isolated in this work is a serovar adapted to the human species, I think it is necessary to better justify this feature.

Introduction

Line 54. Change derivative to derived.

Line 55. Remove "First".

Line 67. Remove in Tolima Region (it is written twice).

Line 70: put a dot after Ibagué city. Reformulated the follow sentence.

Objective

The aim of the study is too long. It should be rewritten focusing only on the objective of the work. "To increase the understanding of Salmonella epidemiology in Tolima region, this study aimed to estimate the prevalence of Salmonella in the poultry farms and to determine the genetic relationship".

Material and Methods.

The authors should add a more detailed description of the poultry farms analysed in this study.

Authors of the manuscript should include the statistical analysis section.

Study population

Please, clarify the study population of the study.

Sample collection

Line 90. Please add the age of the animals.

Line 94. Were water samples collected in bags? Or in sterile pots?

Line 96. Please, add a reference that explains why the environmental samples were taken this way.

Salmonella isolation and identification

Line 108. Feed and water samples were 1:10 pre-enriched in buffer peptone water.

Line 113. Add a reference to the culture media. Why did you use broth Rappaport instead of Semi-Solid?

Line 118. Xylose Lysine Deoxycholate in capital letters.

Line 120. Xylose Lysine Tergitol in capital letters.

Line 125. When at least one sample was positive in the bacteriological culture.

Line 127. Molecular confirmation of Salmonella

I do not clearly understand why this technique was performed. The ISO is the gold standard and official method for Salmonella isolation.

Epidemiological survey

I strongly recommend including a table with the parameters and values that have been assessed in the survey study.

Line 184. A sixty-question survey questionnaire was conducted to the farm manager from each studied farm.

Results

Prevalence of Salmonella.

I suggest including a Table summarising Salmonella prevalence results.

Did you find significant differences between samples?

Line 196. I recommend rewriting the first lines of the results rection. A total of 15 farms have been studied from which 4 were found positive to Salmonella spp. (26.67%, 4/15). From all samples analysed, 17.78% were positive for Salmonella isolation (24/135).

Line 201-202. Please remove this sentence (Discussion).

Line 202-204. Please remove this sentence (Material and Methods).

Salmonella serovars.

Due to all serovars isolated were the same, I suggest removing the table.

Risk factors in broiler farms.

Please, homogenize the title of the section according to the material and methods. Besides, follow the order of how the sections have been described in the material and methods (Epidemiology survey was the last). 

Discussion

I strongly recommend improving the discussion. 

Lines 256-257. Please, put a dot after Tolima region – Colombia. These results are in accordance with 27% prevalence reported in chicken meat by the national retail independent stores.

Line 257. Please remove “the” from the 27%

Line 258-261. I suggest the authors rewrite this sentence it is too long.

Line 265-267. It doesn’t make sense to include this fact here. Please, remove it or relocate it elsewhere.

Line 269. Change the Word invade to colonise

Line 276. When authors of the manuscript refer to Salmonella Paratyhphi variant as source of infection the animal’s reservoirs, are you referring to animal infection or Salmonella? How can you explain this serovar adapted to the human species has colonized the intestinal tract of broilers? I suggest authors discussing further this point.

279-283. Authors of the manuscript describe that the strains isolated in the manuscript were MDR. However, this part has not been described in the material and method nor included in the results sections. If it has been performed it should be included.

Lines 290-294. Please a dot after occurring. Some authors have reported Salmonella carcase contamination during…

Line 299. Please change “In addition” to “In line”

Line 302. Please, add a dot after environmental samples. These results…

Line 302. Please, change “factors” to “sources”.

Line 316. Please, add a dot after Colombia. These…
